# Exploring the Presence of Humanoid Social Robots at Home and Capturing Human-Robot Interactions with Older Adults: Experiences from Four Case Studies

**DOI:** 10.3390/healthcare11010039

**Published:** 2022-12-22

**Authors:** Angela Y. M. Leung, Ivy Y. Zhao, Shuanglan Lin, Terence K. Lau

**Affiliations:** 1WHO Collaborating Centre for Community Health Services, School of Nursing, The Hong Kong Polytechnic University, Hong Kong 999077, China; 2Research Institute of Smart Aging (RISA), The Hong Kong Polytechnic University, Hong Kong 999077, China

**Keywords:** humanoid social robot, older adults, living alone, loneliness, social isolation, home-based

## Abstract

Background: Social robots have the potential to bring benefits to aged care. However, it is uncertain whether placing these robots in older people’s home is acceptable and whether human-robot interactions would occur or not. Methods: Four case studies were conducted to understand the experiences of older adults and family caregivers when humanoid social robot *Ka Ka* was placed in homes for two weeks. Results: Four older adults and three family caregivers were involved. Older adults interacted with the social robot *Ka Ka* every day during the study period. ‘Talking to *Ka Ka*’, ‘listening to music’, ‘using the calendar reminder’, and ‘listening to the weather report’ were the most commonly used features. Qualitative data reported the strengths of *Ka Ka*, such as providing emotional support to older adults living alone, diversifying their daily activities, and enhancing family relationships. The voice from *Ka Ka* (female, soft, and pleasing to the ear) was considered as ‘bringing a pleasant feeling’ to older adults. Conclusions: In order to support aging-in-place and fill the gaps of the intensified shortage of health and social manpower, it is of prime importance to develop reliable and age-friendly AI-based robotic services that meet the needs and preferences of older adults and caregivers.

## 1. Introduction

The global population is rapidly aging. In Hong Kong, 2.37 million (31.1%) people will be aged 65 or over in 2036 [1]. In 2016, about 150,000 older adults in Hong Kong lived alone, and almost 300,000 lived with their older spouse [2]. Living alone or with a spouse (who is also an older adult) may exacerbate loneliness in old age [3]. Moreover, 75% of older adults and 45% of soon-to-be-aged (aged 45 to 64) people in Hong Kong were suffering from one or more chronic diseases [4]. It is well reported that older adults are vulnerable to loneliness due to health deterioration and age-related losses, which prevent them from participating in social activities and engaging in interpersonal relationships [5]. 

In the concept of healthy aging, person-centered holistic care highlights the importance of enabling older people and their significant others (e.g., older spouses or other family caregivers) to establish healthy relationships so as to improve older adults’ physical and psychosocial wellbeing [6]. Daily communication within the family and with friends is crucial for older people and this is impactful on their psychological and mental health [6]. Given the complexity of providing care to older adults living alone at home, which typically involves family companionship or social connection, personalized services such as healthy lifestyle promotion, chronic disease management, and adopting and integrating robotic technology in home-based aging care, seems to be reasonable and potentially benefits the older people [7]. Researchers have been exploring ways to utilize different robotic technologies to help and provide care for older adults, particularly socially-assistive robots, which are used to assist older adults with specific physical tasks and meet social and psychological needs [8,9]. Social robots are designed to interact autonomously with people through various application features which adopt the same repertoire of social signals used by humans [10]. Studies suggest that social robots contribute to the reduction of loneliness and social isolation in old age and become a kind of social capital in their homes [11,12]. For example, the Japanese-made baby seal shape *Paro* was tested in cognitively impaired individuals and was found to significantly reduce negative emotions and behavioral symptoms, and improve social engagement and the quality of life of older participants [9,13,14].

There were numerous clinical trials on the application of robots in aged care, in particular for people with dementia and those who are living in residential care facilities [15]. Research on home-based robotic support is very limited. Robotic pets were tested among community-dwelling older adults in New York and suggested that it could be an effective solution for alleviating loneliness in older adults, especially among those who live alone, have fewer social connections, and live less active lifestyles [16]. A recent study showed that older Chinese adults were highly interested in having social robots during dining and entertainment in home-based aged care [17]. However, more evidence is needed to examine the acceptance and feasibility of a social robot at home [18]. Especially whether older adults, their spouses, or other family caregivers at their homes accept, use, and even live with a robot. Their experiences and perspectives on using robots are important to determine the use of technology in supporting ageing-in-place. Therefore, the aim of these case studies was to understand the experiences and perspectives of older adults and their family caregivers on using a humanoid social robot over a two week period at home and to explore the potential benefits and barriers of human-robot interactions at home.

## 2. Materials and Methods

### 2.1. Description of the Humanoid Social Robots

The main platform utilized in this study was the robotic-mediated interactive system embedded in a humanoid social robot called *Ka Ka*, which is around 30 cm tall, 20 cm wide, and weighted 2 kg (Figure 1). This robot is designed in a way that it simulates human-robot interactions with a human voice from the robot responding to a human’s verbal instructions or initiating conversations by the robot to a human at a specific time. This cultivates an environment for human-robot interactions and human-human interactions via the robot. There are many features in this robot. However, for the sake of testing the acceptance of the robots by older people, we minimize the features of the robot in this study. Four features were selected based on the findings of the literature review as well as the comments from the potential users of the robot (the community-dwelling older adults): (1) interactive modalities, (2) calendar planning and task reminder, (3) promoting healthy lifestyle, and (4) puzzle games. The interactive modalities allow older people to listen to music, stories, weather reports, and the news, and this is initiated by voice commands (that is, older people instruct the robots to turn on the music players by saying ‘*Ka Ka*, listen to Teresa Teng [a pop singer who sang Chinese songs in 1970s]’. Using the feature ‘calendar planning and task reminder’, older adults received voice reminders from the robot for the scheduled important events (such as meeting friends on 12 September 2022 at 10 am) and health-related activities (such as taking medicines at 11am, drinking water, and doing exercise). Moreover, schedules can be set up in the robots to guide older adults when and how to work out (doing stretching exercise). The robot also provides puzzle games and quizzes for older adults to stimulate and practice their cognitive functioning.

### 2.2. Recruitment

Purposive sampling was adopted in study recruitment. Advertisements (in Chinese) were distributed in media and community elderly centers. Those who were interested in participating in the study were directed to contact the study team. People with different genders, ages, educational backgrounds, and occupations were included to reflect a diversity of views. The inclusion criteria for older adults were (1) ethnic Chinese aged 60 years and above, (2) need home-based care/assistance from family caregivers, including spouse, (3) live with a family caregiver in the same household or in another household but with regular contacts (at least once per week), and (4) are able to communicate in Cantonese or Putonghua. The exclusion criteria were those with (1) acute mental illness (such as PTSD) or severe physical disabilities (such as cerebral palsy), and (2) those who have limited access to electricity and no Wi-Fi at home. HKD$200 (~U$26) supermarket shopping vouchers were provided to each participant as incentives to compensate the time they spent on interviews after using the robot.

### 2.3. Ethical Approval

We obtained ethics approval from the Human Subjects Ethics Sub-committee of the [anonymized] University (Reference No. HSEARS20220113001). An information sheet was provided to participants and written consent was gained before commencing data collection. The purpose and methods used in the study were explained to them. Pseudonyms were used in the reporting of findings to protect the confidentiality of participants.

### 2.4. Robot Set-Up

For the sake of personalized the features of the robot, we collected information about the preference of the user (older adult). For example, the older adult’s profile, favorite songs, and important events (including the time to take medicines and the date for next medical appointment). All information was installed into the robots through the *Aged Care* app (see Figure 2). To increase the affiliation with the robot, we introduce the robots to the older adults/family caregivers by naming *‘Ka Ka’* which is a common name in Chinese society. Older adults/family caregivers were encouraged to update and manage the calendar reminder by themselves. When setting up *Ka Ka* at home, we carefully assessed the environment, older adults’ daily activities and preferences, and we identified a safe and comfortable place together with the older adult/family caregiver to make *Ka Ka* visible and accessible to the users.

### 2.5. Train Older Adults and Family Caregivers 

We provided on-site training and showed older adults and family caregivers how to operate *Ka Ka* through verbal commands and how to update the calendar through the *Aged Care* app. We set up a technical support hotline so that older adults can easily contact the technical support team at any time. If necessary, the technical support team paid home visits to solve the technical issues. 

### 2.6. Data Collection 

We encouraged each older adult engage with different features of *Ka Ka* every day for 2 weeks. Login frequency and duration of the human-robot interaction was collected in the backend management system. We utilized the data to examine older adults’ engagement with the robot and preference of specific features. Upon completion of the 2-week trial, we conducted individual face-to-face interviews with the older adults at their home and telephone-interviews with the family caregivers. We selected a qualitative descriptive methodology, which enabled participants to describe their experiences of engaging with *Ka Ka* in their own language [19]. Each interview lasted for approximately 20 min and all of them were digitally recorded. Semi-structured interview guidelines for older adults and family caregivers were developed and derived from the literature (Table 1). 

### 2.7. Data Analysis

Descriptive statistics were utilized to summarize the sociodemographic characteristics of older adults and the human-robot interaction. Descriptive data were presented in frequency (n) and percentage (%). Interviews were transcribed verbatim by a trained research assistant and independently checked by the project team. Thematic coding and content analysis were undertaken independently by two project team members to inductively explore older adults and family caregivers’ experiences of engaging with the robot *Ka Ka* at home [20]. An audit of 15% of the coded segments was performed, discussions were made, and agreements were reached.

### 2.8. Rigor

Rigor was established through credibility, confirmability, and transferability [20]. The interviewers were young adults, trained registered nurses, familiar with *Ka Ka*, and situated themselves as outsiders throughout the interview sessions. To achieve credibility, they bracketed themselves and did not assume how acceptable or user-friendly *Ka Ka* was and held frequent debriefing meetings and encouraged peer scrutiny to reflect on their positioning [21]. Secondly, data sources triangulation (collected data form older adults and their family caregivers) were carried out to gain multiple perspectives in the study [22]. Confirmability was established by recording coding and further supporting quotations from participants for each theme. Transferability was achieved by providing a detailed study design and process to enable readers to understand the findings from the dataset.

## 3. Results

### 3.1. Case Information

Four older adults completed this 2-week trial. Case 1 was an 81-year-old woman who lived alone and independently at home. She spent most of her time at home by herself. Her favourite things to do every day at home were to watch TV and listen to a radio station. Her daughter, who lived in another flat in the same neighbourhood, visited her every day. She was concerned about changes in her mother’s memory and the lack of external stimulus to her which may have impacts on brain activities. Case 2 was a 76-year-old woman, living by herself at home. Her husband lived in a local nursing home, but she has not seen him in person for a long time due to the COVID-19 visitor restriction policy. She participated in community and church group activities actively. She was well-connected with her daughter, who provided regular visits and assistance with domestic matters. Case 3 was an 86-year-old woman living alone in a retirement village. She had worked and lived in the United States before she retired. She sometimes joined activities arranged by the retirement village. She showed her worries about using a laptop and the internet. Case 4 was a 65-year-old man living with his spouse. This couple support and look after each other. They used to have a dog as a pet but they lost it. They still missed the dog’s companion and kept its photos at home. Due to the social distancing of the COVID-19 pandemic, they mostly stayed at home. See Table 2 for the detailed demographic information of the participants.

### 3.2. Interaction with the Robot in Two Weeks

We collected data on the engagement with *Ka Ka* by the participants (see Figure 3). Records from the robotic system showed that older adults used the features of *Ka Ka* every day, but the frequency of human-robot interaction varied, ranging from 2 to 26 times per older adult per day. A total of 506 human-robot interactions were recorded in these two weeks (with an average of 126 interactions per older adult). There was a significant increase in human-robot interactions in the last two days of the trial. 

Figure 4 shows the timeslots of participants interacted with *Ka Ka* in a typical day. A general timeframe for human-robot interactions was from 6:00 AM to 8:00 PM. All participants preferred to interact with *Ka Ka* from 2:00 PM to 5:00 PM. Other two peak time slots were from 8:00 AM to 1:00 PM and from 7:00 PM to 8:00 PM, respectively. The majority of the participants stopped interacting with *Ka Ka* after 9:00 PM. 

‘Talking to *Ka Ka’*, ‘listening to music’, calendar reminders, and weather report were the most commonly used features. All participants listened to music via *Ka Ka,* although the frequency varied across the participants. Two participants (50%) listened to the stories and only one watched the videos in *Ka Ka* on how to do physical exercise. Figure 5 shows the frequencies of use in each feature of the robot by the four participants in two weeks.

### 3.3. Qualitative Findings 

Three key benefits and two aspects for improvement were identified in the qualitative interviews. The strengths of *Ka Ka* include providing emotional support to older adults living alone, diversifying older people’s daily activities, and enhancing the dyadic relationship between older adults and family caregivers. 

#### 3.3.1. Providing Emotional Support to Older Adults Living Alone 

The utilization of *Ka Ka* was reported to reduce feelings of loneliness and boredom among older adults living alone. As a family caregiver stated: ‘*having a short conversation with Ka Ka, even just asking Ka Ka to report weather could be a kind of external stimulation for my mother and this probably reduces her feelings of loneliness*’ (Family caregiver 1). Verbal interactions with *Ka Ka* provided older people with a sense of warmth and encouraged older people to live more actively, especially for the older people who did not have any friends and were unwilling to socialize with others or had no hobbies. An older lady expressed her experience as ‘*Ka Ka reminded me to eat food every day, which made me feel so warm. I am happy to follow Ka Ka’s reminder*’ (Case 2). One older man appraised *Ka Ka* as ‘*a family member*’ because it played a companion role in providing daily reminders and talking to him at any time. This benefit was expressed by both older men and his spouse as follows: 

*‘Ka Ka presents an extra voice at our home. We regard Ka Ka as another family member’* (Case 4)*. ‘Ka Ka’s voice was soft and charming, which made us feel comfortable and calmed us. I think the female voice of robot is adorable. The reminder function is helpful for my husband’s decreased memory.’* (Family caregiver 4)

#### 3.3.2. Diversifying the Daily Activities of Older Adults 

To the participants, *Ka Ka* was found to diversify and enrich the lives of older adults who live alone and improve their quality of life. *Ka Ka* as a platform provides different categories of audio and videos, covering music, voice-led physical exercises, voice-led interactive puzzle games, and storytelling. Older adults can ask *Ka Ka* to play a song directly through verbal command, which is considered ‘simple and convenient’. In addition, *Ka Ka* has functions such as reporting daily news and weather forecasts. One older adults described:

*‘I am highly satisfied with the music function, and I used it every day’* (Case 1); *‘I followed the video to do some physical exercises every day. Ka Ka even encouraged me and cheered me up when I was doing it’* (Case 2); and *‘I found that the news reported by Ka Ka was the timely ones, which is useful for me to know what has happened in the community.’* (Family caregiver 4).

#### 3.3.3. Enhancing the Dyadic Relationship between Older Adults and Family Caregivers

Some family caregivers indicated that they received direct benefits from the utilization of and interaction with *Ka Ka*. To them, *Ka Ka* is a channel to enhance the dyadic relationship between older adults and family caregivers. In the interviews, some family caregivers asserted that they worked with their care recipients (older adults) to explore how to interact *Ka Ka* (make Ka Ka react with them), discuss the functions in *Ka Ka*, share their experiences of engagement with *Ka Ka* and find out solutions to issues, if any. The mutual relationship was improved by sharing experiences and having more common languages. As one older adult asserted: 

*‘Ka Ka has amplified my conversations with my wife’* (Case 4). A family caregiver also *shared* that *‘Ka Ka is a new member, and it has become a new and common topic among the family members’* (Family caregiver 4). 

### 3.4. Two Aspects for Improvement in the Design of the Robot

Older adults and family caregivers suggested two main aspects of design that could be improved to enhance the acceptability of robots at home. They raised a connection issue between the robot and tablet, hoping that the robot can be more user-friendly and its response to a human could be enhanced. 

#### 3.4.1. Connection between Robots and Tablets

*Ka Ka* does not have a monitor; therefore, it is connected to a tablet or a laptop so that pictures or videos could be shown to the robot users. Some older adults criticized that such an external device was inconvenient to them. On a few occasions, *Ka Ka* and the laptops/tablets were disconnected, and the older adults were very worried. As the older adults and their family caregivers stated:

*‘The connection with a laptop made it quite hard for me to use it.’* (Case 3)

*‘My mom tried her best to explore the features of robot; however, I felt there were too many technical issues for her to handle.’* (Family caregiver 2)

Technical issues related to the robot seem to be a concern to the participants. Some participants suggested connecting the robot with a phone or web-based application rather than a laptop or tablet, as they felt that they were more familiar with their own mobile phone than a laptop.

#### 3.4.2. Enhancing Robot’s Response to Human

Another suggestion was to enhance *Ka Ka*’s ability to respond to humans’ verbal commands. Older adults expected *Ka Ka* to be ‘*smarter*’ and be able to respond to their questions instantly. Some older adults indicated that the voice volume of *Ka Ka* was not loud enough, and they have difficulties hearing what *Ka Ka* said. Another older adult wished that *Ka Ka* could be moveable in the future so that they can hear *Ka Ka*’s voice in another room.

Older adults wanted *Ka Ka* to help them in their daily lives as if it was a private secretary (e.g., scheduling daily activities for them and reminding them to take medicine), or serve as an encyclopedia where they could easily find information. As two older adults expressed their expectations:

*‘I hope Ka Ka can entertain my life but also help me with household chores, read stories or long novels for me’* (Case 3).

*‘I hope the robot can remind older adults about health and safety matters, such as take medicine, measure blood pressure…, in a timely manner. She could remind me turn off the gas fire and electrical appliances’* (Case 4).

Although older adults expressed their wishes to involve *Ka Ka* in their daily lives, one family caregiver (Case 4) exclaimed that ‘*the feeling of freshness with a robot could last for one week maximum*’.

## 4. Discussion

This study provided evidence of the presence of a humanoid social robot *Ka Ka* at home for four older adults. This contributes to the limited knowledge we have in the field of gerontechnology and social robotics in a home-based setting. We found that older adults interacted with *Ka Ka* every day during the whole study period, continuously engaging with *Ka Ka* as if it is one of the family members at home. Both older adults and family caregivers agreed that *Ka Ka* was a good companion for older adults at home, and they expected some other features that can support older adults living alone or only with their spouses. This finding was consistent with the finding of a cross-sectional study in which 67% of the Chinese immigrants (including older adults) who felt lonely accepted the companionship of robots and considered technology as a way to alleviate loneliness [7]. 

The participants in this study indicated positively with regard to the feasibility and usability of *Ka Ka* in their own homes, which was aligned with a recent scoping review that older adults reported good feasibility and usability of using social robots in care home settings [23]. According to a recent systematic review, acceptance of social robots in healthcare was found to be mixed and can vary considerably in relation to the function and appearance of the robot [24]. The use of artificial intelligent (AI) in social robots is uncommon. Previous studies only reported the use of AI in diagnostic procedures like AI olfactory systems or medical images [25,26]. The current study added our new understanding of the feasibility of using technology to support older adults in real-life settings (their own homes) where no professionals are available. The interaction with the robot solely rests on older adults’ self-initiatives or the pre-set timer when the robot talks to the older adults at specific time (such as reminding them to take medications or take their breakfast). Such interactions are valuable to older adults who live alone by themselves or with their spouses, because the robot act as a member of the family. The personalized feature of *Ka Ka,* such as calendar planning and reminders, was helpful in supporting older adults’ daily home life. It was beyond the function of companion robots (such as *Paro*) in which Paro’s companions were evidenced to reduce older people’s emotional loneliness effectively [11]. As medication adherence and good nutritional intake are two big issues in aged care, *Ka Ka* seems to play an important role in these particular aspects of care.

While identifying various benefits of social robots for therapies or companionships [27], previous studies also pointed out that social robots could never replace human presence, especially their family members [28]. A recent experimental study provided evidence that our brain reacted differently in human-human and human-robot eye contact [29], indicating that humans can easily distinguish communication with social robots from humans. Nonetheless, social robots could be an option when human-being care is not available. The contradiction could be overcome by personalizing the older adult’s real situations.

In this study, the most frequently used features of *Ka Ka* were verbal interactions and playing music. One possible reason was that older people can instruct the robot through verbal commands, which fits older people’s abilities. They found that the feature of verbal interaction and listening to music was easy to access. This was consistent with a previous study that many older people did not prefer text messaging due to their lower writing ability than the younger generation [30]. The interaction of older adults with robots was reported to require effective verbal feedback, which was significantly preferable and increased the engagement with the robot of the older participants [31]. The second reason was that verbal interactions with and remainders from *Ka Ka* played an essential role in providing emotional support to older adults in this study, particularly those living alone at home. The qualitative interview suggested that *Ka Ka* enriched the daily activities of older adults who lived alone and spent most of their time at home watching TV or listening to radio stations. A novel finding of this study was that the robot’s voice (female, soft, and pleasing to the ear) could also bring a pleasant feeling to older adults. An older adult and his family caregiver regarded *Ka Ka* as a family member due to the voice of a ‘*new member*’ presented at home. An emotional tie was built up among them, and they even wanted to keep the robot for a longer time. The finding that *Ka Ka* might positively impact upon the person’s mood is consistent with previous studies concluding that social robot-mediated intervention was promising to improve interaction/engagement and reduce loneliness and depressive symptoms for older adults [15]. The last possible reason was that personalized content in the robot enhanced its usage. For example, the songs were prepared according to the users’ recommendations and preferences. Except for offering some entertainment value, the personalized robot features could create some feelings of being cared for by older adults [23]. 

We have observed that the frequencies of engagement with *Ka Ka* were comparatively high during the first week and the last two days when we contacted older adults to schedule a time to complete the study and return the robot to the research team. Older adults seem to engage with *Ka Ka* in a regular manner, usually in the morning after breakfast and in the afternoon between lunch and dinner. We found that older people were more likely to follow a stable and fixed rhythm of life. They seemed to use features of verbal communication and listening to music more frequently during this time to fill their spare time. Moreover, some older adults showed activity when we contacted them but became passive afterwards in using *Ka Ka*, and this pattern was consistent to previous studies that technology-related fear might inhibit older people to engage with technologies [32,33]. The fear includes the possibility that technology may be hard or impractical to use, the possible negative effects of sensor radio waves on their personal health, and that extra burden may be brought to their adult children [33]. We have observed a fear of breaking the robot as indicated by an older adult at the time when we intend to recruit her to the study. Two older adults in this study expressed their barriers to using the laptop connecting to the robot. In future studies, more attention should be paid to supporting their acceptance through hands-on training and providing a clear user guide to meet their unique personal needs [34].

Hearing ability may affect the use of robots at home. We found that older adults in this study had different requirements on the voice volume of *Ka Ka* due age-related hearing loss problems. A cross-sectional study in Hong Kong reported that about 28% of older adults living in the community had hearing impairments [35]. It suggests that hearing, as well as other sensory function changes, should be considered by robot designers.

Although *Ka Ka* has artificial intelligence, its responses are based on a verbal command database and this can only be enriched by frequent interactions with humans. *Ka Ka* is the first Cantonese-speaking humanoid robot, and this first trial inspires the subsequent establishment of a verbal command database in the Cantonese-speaking community. A further study is needed to examine the research gap in science and inform subsequent robotic development works. 

### Limitations

There are some limitations of this study. First, users’ experiences might have been disturbed by technical issues and Chinese older adults may have had hesitations about contacting the project team for concern of ‘botheriing’ others. The challenges in technology may hinder older adults from using the robots. Second, this sample cannot represent the heterogeneous aging population, as we only have four cases in this study. Further investigation should be made with more people with different age ranges, socioeconomic status, and cognitive levels. Moreover, due to the ‘Hawthorne’ effect [36], participants might modify their behaviours or performance when they are aware that they are being observed. To avoid this effect, establishing rapport and making the subjects feel relaxed in the presence of a participant observer are warranted in future studies. Third, the involvement of family caregivers is limited in this study and cannot represent all caregivers. The current findings should be considered as exemplars. It remains unclear if the results could be applied to other aged populations in Chinese society. Further research is clearly necessary.

## 5. Conclusions

Older adults and family caregivers in this study were open to the use of social robots in their daily lives. In order to support aging-in-place and fill the gaps of intensified shortage of health and social manpower, it is of prime importance to develop reliable and age-friendly robotic services that are tailored based on the needs and preferences of an older adult. 

## Figures and Tables

**Figure 1 healthcare-11-00039-f001:**
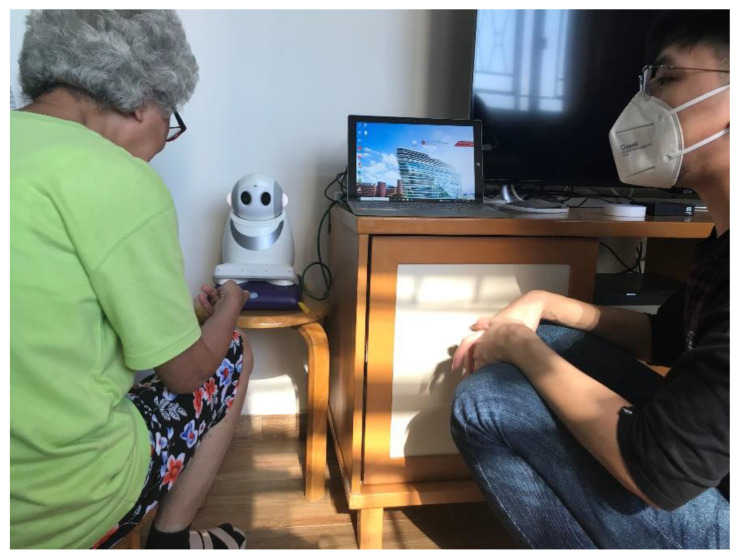
The participant was engaging with *Ka Ka* at home.

**Figure 2 healthcare-11-00039-f002:**
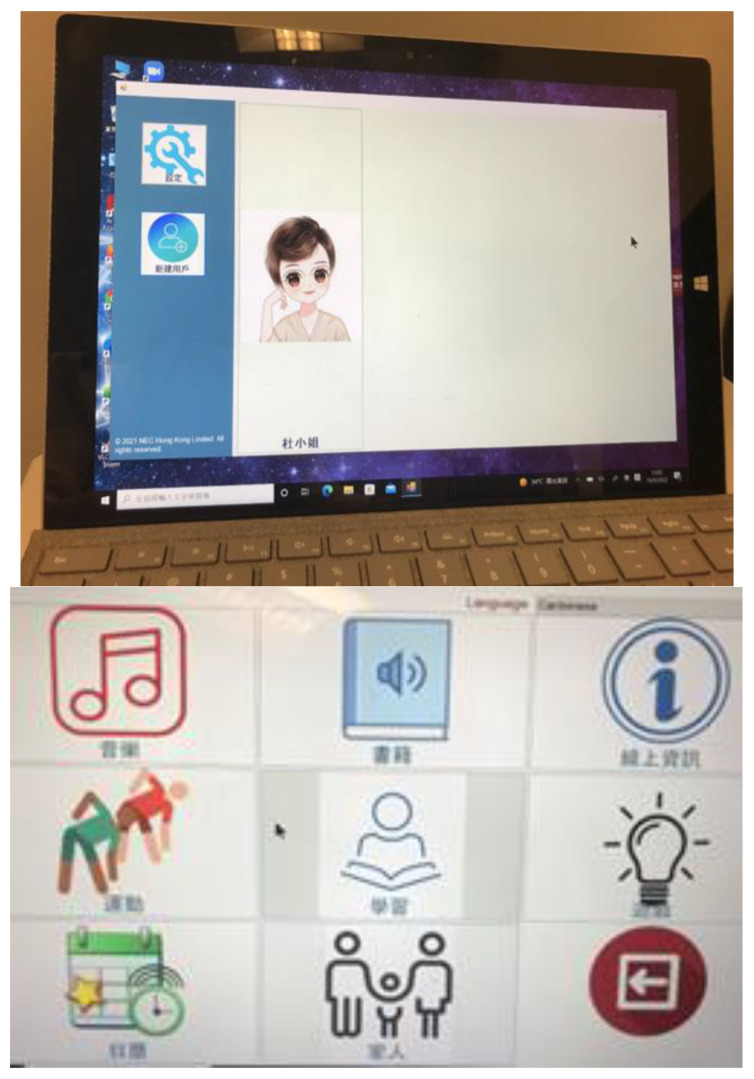
The *Aged Care* app captures the older person’s preferences in music, books, news reading, physical activities and the family members’ contacts.

**Figure 3 healthcare-11-00039-f003:**
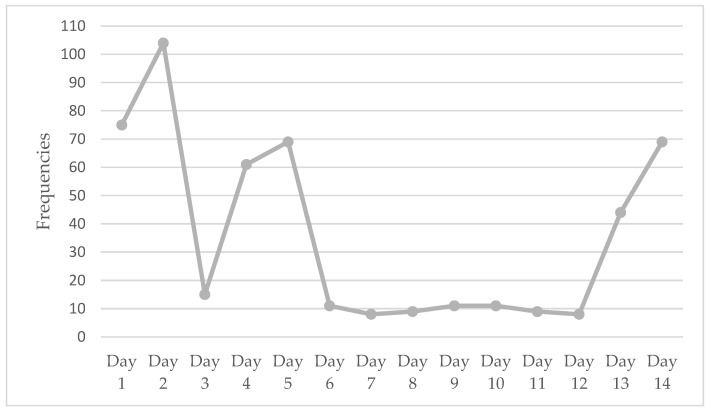
Frequencies of human-robot interactions in two weeks.

**Figure 4 healthcare-11-00039-f004:**
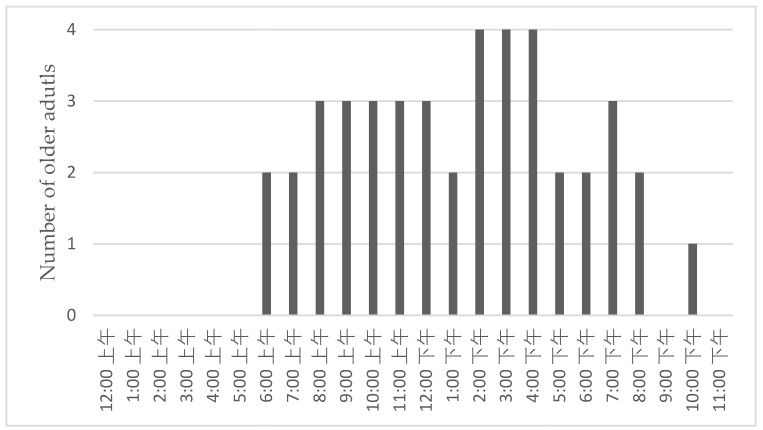
Older adults’ interaction with *Ka Ka* during a typical day.

**Figure 5 healthcare-11-00039-f005:**
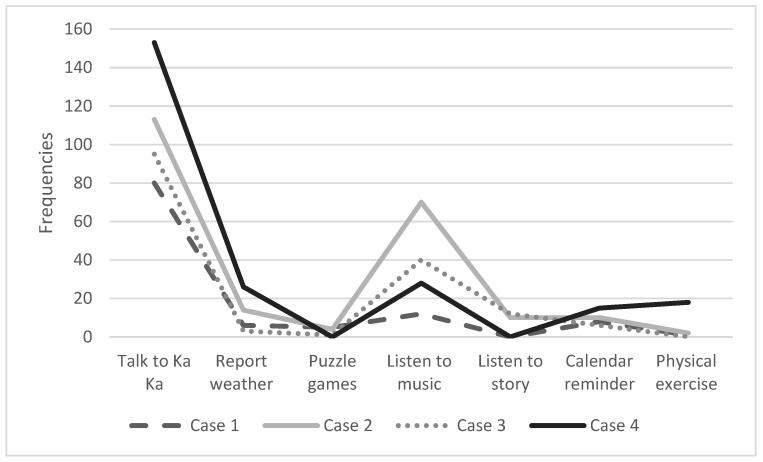
The use of different features by the four older adults over two weeks.

**Table 1 healthcare-11-00039-t001:** Interview guide for older adults and family caregivers.

No.	Interview Guide for Older Adults
1	What is your experience when the robot is placed at your home? (can you tell me how you think about this robot, which feature do you think is the most helpful and which one is not very helpful)?
2	What do you think about the acceptability or satisfaction of using the robot at home?
3	Do you think the robot can be used as part of your daily life activities? How does the robot help you in daily life?
4	Do you think the robot is easy to use? What are the difficulties when you are using it?
5	Have you tried to use the robot to contact other people? How do you feel about it? Do you think you would like to keep this robot at home in the future (shall we give this kind of robot to other older people living by themselves)?
**No.**	**Interview Guide for Family Caregivers**
1	What do you observe when the robot is placed at your mum’s/dad’s home?
2	Which features do you consider as helpful to your mum/dad/spouse?
3	Did the robot help to promoteyour relationship with your mum/dad/spouse?
4	What other features should be included into this kind of robot?
5	Do you think the robot is suitable for older people living alone at home?

**Table 2 healthcare-11-00039-t002:** The demographics of the participants.

	Case 1	Case 2	Case 3	Case 4
Age	81	76	86	65
Gender	Female	Female	Female	Male
Living status	Living alone	Living alone	Living alone	Living with spouse
Occupation	Retired	Retired	Retired	Retired
Marital status	Widowed	Married	Widowed	Married
Education level	Primary	Primary	Secondary	Secondary

## Data Availability

Data is available per requisition.

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
