# Peer review of "Exploring the Presence of Humanoid Social Robots at Home and Capturing Human-Robot Interactions with Older Adults: Experiences from Four Case Studies"

_healthcare, 2022, doi:10.3390/healthcare11010039_

Round 1

Reviewer 1 Report

Title: As this is a qualitative study, I suggest changing "Evidence from Four Case Studies", a more quantitative term, to "Experience from Four Case Studies", which is more consistent with the methodology used.

The topic of study of this paper is very timely given the ageing situation of the world's population. 

The introduction manages to approach the topic of study adequately by providing the social context of ageing and some experiences with the use of robots and the elderly population.

The use of a qualitative methodology is very timely. All the details of the methodology used are well detailed.

I consider the number of informants to be small. When presenting the "case study" research, I would advise to go into more detail on each of the four older adults participating in the study and then go on to present the three key benefits and two aspects for improvement.

The discussion, limitations and conclusions are very appropriate.

I encourage the authors to improve this paper for publication as it is of great interest.

Reviewer 2 Report

This paper is relevant as it deals with a topic of interest such as the presence of humanities social robots in the home and captures human-robot interactions with older adults. Furthermore, this recent study adds interest by addressing this topic through four case studies that sought to understand the experiences of older adults and family caregivers over a two-week period. The findings of this research suggest that older adults interacted with the social robot every day during the study period. The most frequently used functions were: talking to the robot, listening to music, using the calendar reminder, and listening to the weather report. The qualitative results also showed that there are several strengths of using these robots, such as: providing emotional support to older adults living alone, diversifying their daily activities and improving family care. 

The title and abstract of the manuscript closely correspond to the content of the article. 

The main strengths of the manuscript are related to the social interest in the chosen topic. On the other hand, the theoretical framework of the study presents an interesting and wide-ranging review of primary and secondary scientific sources of interest. Likewise, the document succeeds in addressing a good number of references and studies on this subject (25), of which 23 correspond to current publications published in the last five years. 

In relation to the method section, it would be desirable for the authors to include the procedure for recruiting participants. Also, it should be clarified whether the study participants received any kind of reward for their participation in the research. 

Furthermore, the authors should substantiate the qualitative methodological procedure used. Thus, the authors should make it clear whether the research method used was deductive or inductive and, above all, what theory underlies the methodological development they used? On the other hand, I wonder if it would be possible for the authors to provide any results on the reliability of the qualitative analysis.

Reviewer 3 Report

The article is interesting and valuable. Work is of great importance for an aging social group. The introduction state the purpose of the paper. The work is important for collegoues working in the field of medical robotics.The work is thematically coherent. No errors in reasoning were noted.The work is scientifically credible. The correct mindset was adopted when writing the article. The number of references is proper and references are modern.The charts have the correct aesthetics. The number of literature references on the basis of which the work was created is sufficient and the references are modern. The work has a significant practical value.

What opportunities does a social mobile robot offer in relation to a standing one connected to a computer?

Major:

Authors should articulate the research gap found in science.

The biggest shortcoming of the article is the lack of mathematical formulas.

In subsequent scientific works, the research group should be enlarged.

Conclusions section is too short.

Minor:

You should elongate sections 2.2, 2.3, 2.5, 2.6, 3.3, 3.4 etc. or re-edit all text.

You should reword the sentence: ,,Figure 4 shows" etc.

There are editorial errors in the article for example: ,,how do you think...". You should check all the text carefully.

You should remove the word technology etc. from keywords.

Work is moving in the right direction.

Reviewer 4 Report

The manuscript could benefit from some improvements:

·       Develop a critical discussion of the limitation of using robots to replace manpower in  caregiving and interactions with older adults

·       Indicate also the level of acceptance of AI and AI-based robots at the societal level in Hong Kong and in other countries. What are people's expectations, acceptance, and what type of fears people have when talking about AI-robotic-based services in their homes

·       And about older people, are those expectations, acceptances, and fears different from the rest of the population?

·       Did participants receive any financial incentives to participate in the trial? Or any type of incentives? If not, is there any evidence of the role of incentives in such trials?

·       Also discuss the setting. There are numerous situations in which people modified their behavior as a result of being observed or part of a trial (the so-called Hawthorne effect. Address such limitations here, for your study and suggest ways to prevent such an effect. 

Round 2

Reviewer 2 Report

I consider that the authors have made significant improvements in the content of the manuscript. I endorse the article.